# Research on Affective Interaction in Mini Public Transport Based on IPA-FMEA

**Qizhao Peng, Weiwei Wang \***  **, Xiaoyan Yang \*, Yi Wang and Jian Chen**

School of Design and Art, Shaanxi University of Science and Technology, Xi'an 710021, China
\* Correspondence: wangweiwei@sust.edu.cn (W.W.); yangxiaoyan@sust.edu.cn (X.Y.)

**Abstract:** In the promotion of sustainable modes of transport, especially public transport, reasonable failure risk assessment at the critical moment in the process of service provider touch with users can improve the service quality to a certain extent. This study presents a product service touch point evaluation approach based on the importance–performance analysis (IPA) of user and failure mode and effect analysis (FMEA). Firstly, the authors capture service product service touch points in the process of user interaction with the product by observing the user behavior in a speculative design experiment, and perform the correlation analysis of the service product service touch point. Second, the authors use the IPA analysis method to evaluate and classify the product service touch points and identify the key product service touch points. Thirdly, the authors propose to analyze the failure of key product service touch points based on user-perceived affective interaction and clarify the priority of each key touch point. Finally, reluctant interpersonal communication, as the key failure caused by high risk, is derived according to the evaluation report, which leads to establishing new product service touch points and improving the overall user experience to promote sustainable transports with similar forms and characteristics.

**Keywords:** product service touch point; importance performance analysis; failure mode and effect analysis; affective interaction

## 1. Introduction

The product–service system (PSS) design, motivated to fulfill the commands of the user, is regarded as a useful strategy to face current comprehensive development issues [1]. One of the trends of the PSS design is sustainability, which shows the great potential contribution that a strategic design approach can make to stimulating and supporting societal embedding [2]. The principle of sustainable development emphasizes the important role of public transportation: reducing empty seats in transportation can help reduce the consumption of energy, emit less greenhouse gases, and cause less pollution [3–6]. Regardless of the mode of transport, improving its utilization and efficiency is a sustainable initiative with evidence [3,7]. Vehicle miles travelled, a criterion, is defined as the miles of traveling vehicles required to meet commuting needs during a given period at the macro level [8]. All other conditions being stable, if this figure can be reduced, the traffic efficiency will increase, which equals less traffic congestion to occur and better environmental impact [9]. One of the best approaches is making full use of the seats of the vehicle in principle, and encouraging and normalizing sharing transport in practice [10]. After reviewing related research, mini public transport, such as mini buses and ridesharing, is regarded as the most balanced transport between sustainability and efficiency [11–13].

At present, the largest market for ridesharing is Europe (Mordor Intelligence, GLOBAL RIDESHARING MARKET—GROWTH, TRENDS, COVID-19 IMPACT, AND FORECASTS (2023–2028), <https://www.mordorintelligence.com/industry-reports/ridesharing-market>, [accessed on 4 April 2023].). The Global ridesharing market is expected to grow at a CAGR of 18% over the forecast period from 2022 to 2027, which shows a prosperous prospect. Due

to the COVID-19 impact, although ridesharing has witnessed massive declines in demand, many believe the ridesharing market can emerge again (Ibid); therefore, it is not too late and still economical to promote ridesharing. Didi Chuxing, BlaBlaCar, Lyft, Uber, and Zimride provide various applications all over the world but similar service processes, which is the specific application scenario of this research. However, two main factors are limiting the development of mini public transport: technological factors and the passenger experience of the service [12]. These companies continue to optimize the service experience of their products, but the overall experience of users has not changed but only some details have been modified due to such great behavioral inertia, which makes it difficult to detect the starting point. Grounded in service design and interaction design, the service experience of public transport is the focus of this research.

To examine the main factors affecting the service experience of public transport passengers, a speculative design experiment by postgraduate students from the Glasgow School of Art produced a kind of driverless ridesharing vehicle prototype. Additionally, a speculative commuting ridesharing service process was proposed. Speculative design is a critical design experiment based on a virtual prototype that provides an idealized experiment object and scenario for a complex social issue [14]. Many of the large-scale speculative design experiments have yielded good results and realized some social benefits [15–17]. A report has even illustrated that the exploration of speculative design as a participatory approach to more inclusive policy identification and development in Malaysia is of evident practice meaning [17]. The speculative design is becoming a future-oriented method in theoretical design methodology to resolve complex issues and rethink the present through product design [18].

Based on the initial speculative prototype, during the study of the service users, many details that could be improved were discovered, which are comprehensive and numerous. In the face of these seemingly illogical and unsystematic details, a reasonable scientific system of analysis that can assess the importance and priorities between these various points plays the most important role. In other words, the complexity of identifying key research object units of value from complex objects and locating key items of value from complex service processes reflects the necessity for this research [19]. Therefore, the core aim of this research is to identify the main contradiction in the service process of public ridesharing and to capture the main aspects of the contradictions. Specifically, the aim is to examine the main factors affecting the passenger experience of mini public transport services and further analyze their manifestations, main causes, possible consequences, and risks of failure based on which recommendations for improvement and optimization are made to avoid the creation of pain points.

After reviewing related studies and drawing on effective methods, based on the core aim of this research, three main research methods were employed in this research: service touch point analysis, importance–performance analysis (IPA) of the user, and failure mode and effect analysis (FMEA). Service touch point analysis stages modularize and detail complex processes so that the object of study is transformed from a service process into individual points. IPA analysis quantitatively evaluates the importance and performance of each touch point to select the key touch points most deserving of focus [20]. Furthermore, FMEA, another quantitative method, analyzes the failure risk of key touch points, and those of high failure risk are supposed to be redesigned [21]. The system based on the above methods can be called IPA-FMEA, which points out the key aspects through a two-tier evaluation model. IPA-FMEA, as a kind of core-oriented assessment method, is introduced into this research, whose result will be of great guiding benefit to lead the direction of redesign and optimization.

Initial secondary research has shown that affective interactions have a significant impact on the quality of product service. Emotions are compelling human experiences and product service designers can take advantage of this by conceptualizing emotion-engendering products to promote in the market [22]. In addition to the intended functionality of the product, its affective properties have emerged as important evaluation criteria

for the successful marketing of the product [23]. Therefore, an effective way to promote public transport is to improve the service experience of passengers by providing a quality space for affective interaction. Furthermore, emotional factors are likely causes of service unit failure, and it is necessary to introduce the analysis of affective interaction variables in the IPA-FMEA.

The rest of the paper is arranged as follows. The related works are shown in Section 2. In Section 3, the research scenario and method are mainly discussed, including identifying service stages and touch points, identifying key touch points by IPA, identifying failure risk of key touch points by FMEA, and clarifying the priority of touch points with high risk. In Section 4, the research result is described. Finally, a discussion of future work is proposed in Section 5, and the conclusion is presented in Section 6.

## 2. Literature Review

To identify research-worthy priorities from complex service processes, the three research methods, product service touch point analysis, IPA, and FMEA, presented in the relevant literature are highly informative.

### 2.1. Product Service Touch Point

The first task in analyzing complex processes is to simplify and modularize complex objects as much as possible. The product service touch point is a frequently cited method in the literature for classifying services by stage. It is generally accepted that product service touch points are widely present in the service process as the service recipient interacts with the product, the environment, the service, and the communication elements [24,25]. The touch points, as the basic elements of product services, make up these complex service systems whether they are linear, cyclical, or tree-like service processes. It is necessary for the designer to understand the process of service delivery to enable the design of product service touch points in greater depth [26]. One case follows the principles of experience design and uses a list of touch points to develop the concept of early mental health prevention and treatment through experimentation, focusing on an innovative built environment [27]. Another case uses product service touch points as an opportunity to expand the design perspective and embed smart technologies with conductive effects in the smart shirt design process [28].

From the existing research findings, touch point analysis is the basis and prerequisite for service design. Listing service touch points is the first step in analyzing a service process because it simplifies and systematizes the complex object, especially analyzing complex service scenarios with multiple users, multiple devices, and multiple interactions [29]. Ridesharing in mini public transport is a scenario that may involve many passengers, some products, and various interaction approaches. Thus, service touch points analysis is the practical application of this research.

However, touch points are the basic units of the service process, which needs to be evaluated. Additionally, many sources show that there is a large number of product service touch points in a service process. Thus, it is still necessary to refer to the relevant literature and to use appropriate methods for distinguishing vital ones, including IPA and FMEA.

### 2.2. Application of IPA and FMEA

To select some key touch points from a wide range, it is of great benefit to analyze the importance and performance of all product service touch points involved in the service process, and classify them, which is called importance–performance analysis (IPA), a method of analyzing the customer performance of products or services [30,31]. A case used IPA to explore the views of patients and nurses on the priority of rehabilitation nursing service. The IPA matrix was used to show the differences between patients and nurses in the priority of the nursing service, which provided new ideas for nursing service designs [32]. IPA is a simple and effective method that can provide decisionmakers with the index bias that affects the attention of user performance.

The results following the IPA selection are still simplistic, as the evaluation metrics only take into account the subjective perceptions of importance and performance, but ignore the objective fact of whether the touch points are prone to failure or not. Accordingly, failure analysis of key touch points is a necessary supplement. Failure mode and effect analysis (FMEA) is a systematic and forward-looking analysis tool, which is generally used to identify potential risks and safety hazards and remove problems, errors, and potential risks in the system, design, process, or service [33].

FMEA is a failure-oriented analyzing method. A case used FMEA to assess the potential failure of a medium-sized urban hospital and improve the safety of blood transfusion. The research design and method use the probability of occurrence, the severity of the impact, and the detection probability to evaluate each failure mode [33]. Furthermore, FMEA can be an endpoint research method after other analyses. An article proposes that quality function deployment (QFD) technology is used to transform customer requirements into service technology, and the priority of service requirements improvement is determined by combining FMEA. This method uses QFD and FMEA to design a local pension policy that meets the needs of the elderly, provides clear design, improves service quality, and helps to establish a local aging policy [34].

IPA and FMEA are two assessment methods with different criteria: IPA can filter out the more important points and can be used as the primary assessment; FMEA can determine the risk value of each point and can be used as the end assessment. A two-tier assessment method based on these two methods is known as IPA-FMEA.

### 2.3. IPA-FMEA

IPA-FMEA contains two phases, which are widely used in guiding PSS design. A case combined IPA and FMEA to evaluate user satisfaction to improve the service quality and effectiveness of a company [35]. Another case used IPA-FMEA to optimize clothing industry product design to achieve higher profitability, more environmental benefits, and social effects. [36] The examples in the literature show that IPA-FMEA can, to a certain extent, pinpoint important, low-performing, and regular objects with a high risk of failure from a complex of components. As for ridesharing in mini public transport, it is a universal, multifaceted, linear service process. Accordingly, IPA-FMEA can meet the basic needs for studying mini public transport ridesharing.

However, there are some shortcomings shown in previous research. Most cases simply take two traditional approaches and combine them in a crude way, so that it is necessary to carry out some innovations for these two methods. The traditional IPA method generally uses the average value of user importance and performance value as the classification condition in the analysis process, and the perception and attitude of different people are fuzzy and uncertain [20,32,37]. Therefore, based on the traditional IPA method, this study introduces the standard score as the classification condition of the product service touch point index to improve objectivity. The traditional FMEA method only takes mechanical failure into consideration. However, it does not take affective failure into consideration. Based on traditional FMEA, the measurement of the user tolerance area is introduced, considering the gap between the ideal service value and expected service value, which analyzes the failure risk of key product service touch points.

### 2.4. Preferred Method Discussion

In contrast to IPA-FMEA, there are several other assessment methods that can be used as pre-studies for service design, but these methods are more or less inadequate for this research topic. For example, a group presented a fuzzy-neural-based IPA (FN-IPA) that integrates fuzzy set theory, back propagation neural network, and three-factor theory. However, this approach is more applicable to dynamic and irregular service systems but not to stable and regular services such as carpooling. There is another example in that a group presented an FMEA method based on fuzzy methodology, which can be used to transform linguistic subjective evaluations into objective values by fuzzification and

defuzzification. However, the number of experts involved in this method of assessment is so small, usually no more than six, that the results of the study depend to a large extent on the evaluators and, as a result, it is poorly represented.

With reference to the above-mentioned literature, the research methodology for this paper was determined to identify specific study subject units through service touch points analysis, and assess them according to the two-tier evaluation system of IPA-FMEA. The concept procedure derived from a combination of three methods in the literature is shown in Figure 1.

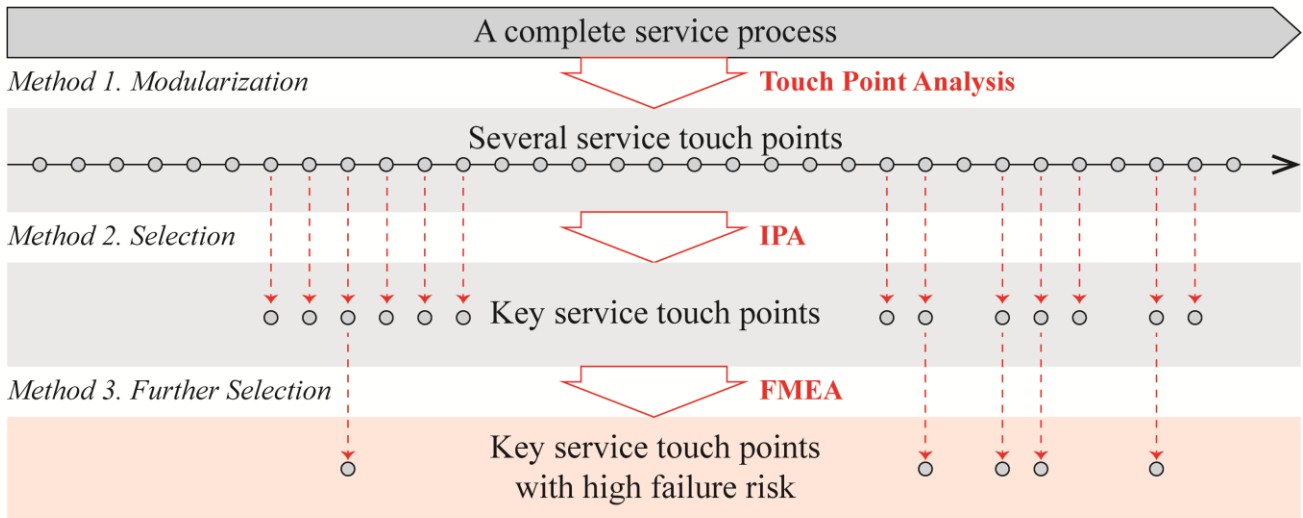

**Figure 1.** Analyzing methods and procedure diagram.

## 3. Research Scenario and Process

### 3.1. Speculative Experiment Scenario

To point out the core factors that make passengers dislike sharing transport and mini public transport, postgraduate students from the Glasgow School of Art held a speculative design experiment about future sustainable transport. Speculative design experiment is a popular design methodology often used to discover the key insights of future social issues [18]. As for this experiment, firstly, the group designed a kind of driverless ridesharing vehicle, as the speculative design experiment scenario. This minibus offers six seats to passengers who do not know each other, which is the basic user research scenario this paper studied, as shown in Figure 2. A variety of materials, equipment, and sites are used to compose the simulation scenario. For instance, cardboard is used to make the shell of the vehicle, and several common sorts of the chairs are used to represent the seats in the vehicle. In addition, halls, stadiums, pavements, and parks were regarded as future virtual environments for experimentation. Students, teachers, passers-by, drivers, and many other people were invited as volunteers to play the roles in those scenarios.

In addition to the physical scenarios, virtual service systems and processes have also been designed with referring to Didi Chuxing and Uber, and devised as shown in Figure 3, which is the basis of the service touch point analysis. Volunteers were asked to participate with sympathized perspectives to act out an immersive experience, including booking, waiting, checking, getting on, sitting in, and arriving. During the experience, it was of great significance to record the flow in detail and pay attention to the emotional and psychological changes of the participants. After the immersive experiment, participants were asked to complete the first questionnaire about the IPA. By inviting as many volunteers as possible or repeating the experiment more times we can obtain more primary research information. The data of the IPA below are based on the results of the questionnaire on the virtual service experience at this time. Additionally, the evidence of the FMEA below are based on observing volunteers experiencing speculative service.

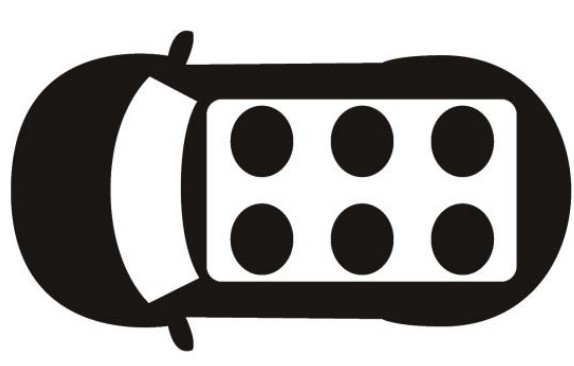
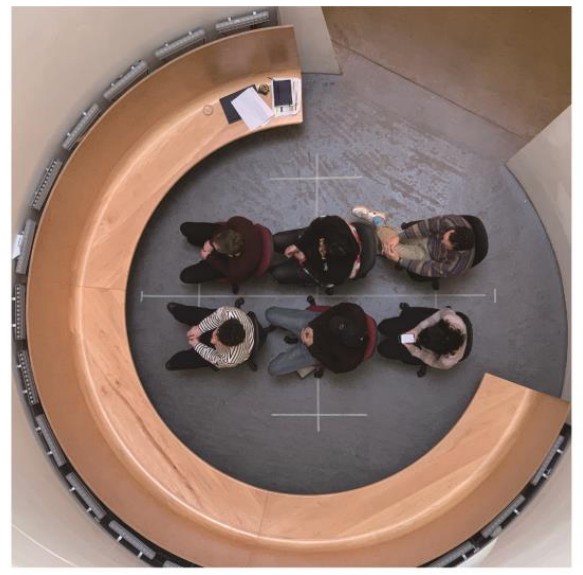

(a)　The Model of Speculative Experiment　　　(b)　The Scenario of Speculative Experiment

**Figure 2.** Speculative design experiment of driverless ridesharing vehicle.

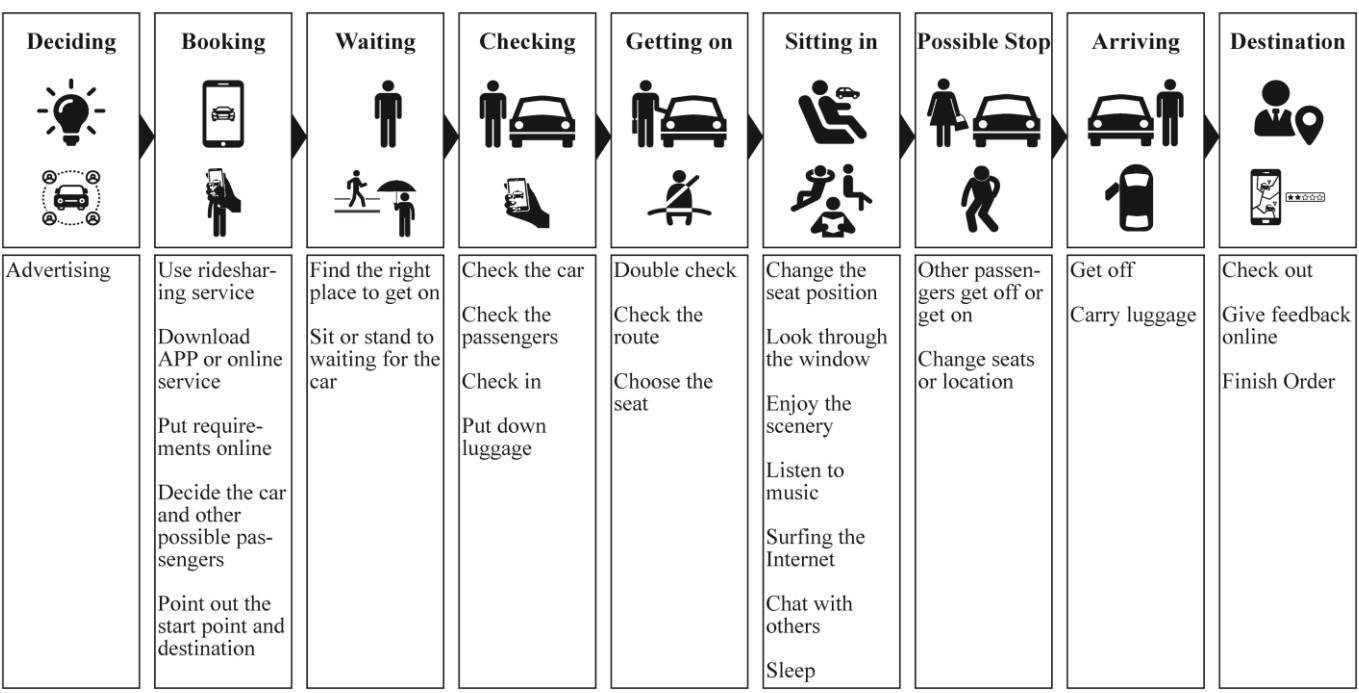

| Deciding | Booking | Waiting | Checking | Getting on | Sitting in | Possible Stop | Arriving | Destination |
|---|---|---|---|---|---|---|---|---|
| Advertising | Use ridesharing service<br><br>Download APP or online service<br><br>Put requirements online<br><br>Decide the car and other possible passengers<br><br>Point out the start point and destination | Find the right place to get on<br><br>Sit or stand to waiting for the car | Check the car<br><br>Check the passengers<br><br>Check in<br><br>Put down luggage | Double check<br><br>Check the route<br><br>Choose the seat | Change the seat position<br><br>Look through the window<br><br>Enjoy the scenery<br><br>Listen to music<br><br>Surfing the Internet<br><br>Chat with others<br><br>Sleep | Other passengers get off or get on<br><br>Change seats or location | Get off<br><br>Carry luggage | Check out<br><br>Give feedback online<br><br>Finish Order |

**Figure 3.** Speculative service process of driverless ridesharing.

### 3.2. Process of Four Phases

In this paper, based on the speculative design experiment above, through the analysis of user behavior identification to build public traffic service touch points in the ridesharing service process, the importance and performance of service touch points are evaluated, and the key touch points are analyzed. The failure analysis model is mainly divided into four stages, as shown in Figure 4. The specific research methods, principles, formulas, and criteria are described in detail later in this section.

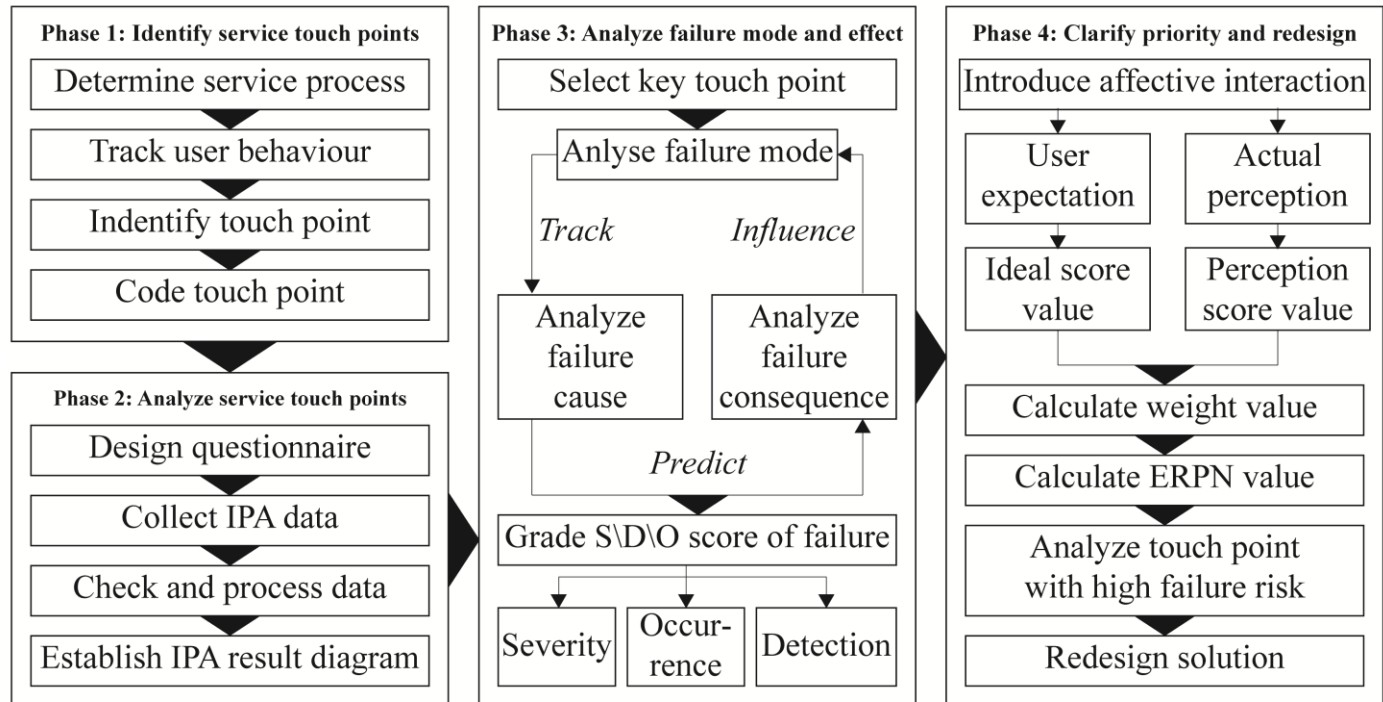

**Figure 4.** Research process consisting of four phases.

### 3.2.1. Identify Service Touch Points

In the first stage, the specific service process is determined, and the product service touch points in the whole process are identified through the analysis of user behavior. The main work in this stage is coding touch points. The product service touch points are coded as T$ij$. Suppose T = {T1, T2, . . . , T$m$} is a specific phase in a specific service process, where T$i$ is the service in phase $i$, $i$ = 1, 2, . . . , $m$. Furthermore, T$ij$ is the $j$th product service touch point in the service process of stage $i$th, the number of sub-product service touch points in each stage is determined according to the specific situation, where $i$ = 1, 2, . . . , $m$; $j$ = 1, 2, . . . , $n$. After coding, the environment and medium of each contact are analyzed.

The coding system in this stage greatly contributes to reducing the complexity of the research procedures and consumed the time of the text work, which makes each service touch point shown in this paper simple and directive.

### 3.2.2. Analyze Service Touch Points

The first step of this stage is designing the questionnaire. All the product service touch points are used as evaluation indexes, and the questionnaire design is carried out for them. The values of importance and performance are integers in the interval between 1 and 5 [38]. The higher the value is, the higher the performance or importance of users at this touch point is. Through the questionnaire, volunteer users were asked to score the performance and importance of the product service touch point according to their own experience [39].

After retrieving the questionnaire, it is imported into the SPSS Statistics 25 software to test the reliability of the collected questionnaire data. The Cronbach's coefficient value is between 0 and 1. The larger the Cronbach's coefficient is, the more reliable the collected data are. If Cronbach's coefficient reaches 0.8, it indicates that the reliability of the scale is good.

The main body of IPA calculates the standard score and plots the coordinates. $A_{pn}$ and $A_{in}$ are the average score values of performance and importance, respectively, for the $n$th touch point, whose calculation is based on Formulas (1) and (2). $S_{in}$ and $S_{pn}$ are the values describing the standard deviation of the importance and performance score from the overall average for the $n$th touch point, whose calculation is based on Formula (3) and

Formula (4). If it is greater than 0, it means that the touch point has an above-average score and if it is less than 0, the touchpoint has a below-average score. The meanings of the other variables in these formulas are as follows: $P_{mn}$ is the rating of the $m$th respondent on the satisfaction of the $n$th touch point, $I_{mn}$ is the rating of the $m$th respondent on the importance of the $n$th touch point; $N$ is the number of valid returns. $MP$ and $MI$ are the overall mean values of performance and importance for all touch points; $SDP_n$ and $SDI_n$ are the standard deviation of the performance and importance scores for the $n$th touch point.

$$A_{pn} = \frac{\sum P_{mn}}{N} \tag{1}$$

$$A_{in} = \frac{\sum I_{mn}}{N} \tag{2}$$

$$S_{pn} = \frac{A_{pn} - MP}{SDP_n} \tag{3}$$

$$S_{in} = \frac{A_{in} - MI}{SDI_n} \tag{4}$$

Based on the magnitude of $S_{pn}$ and $S_{in}$ in relation to 0, the service touch point indicators are divided into four categories [40], as shown in Figure 5. Those with a high degree of importance and performance belong to good work, marked as I; those with a low degree of importance and high degree of performance belong to a possible overkill, marked as II; those with a low degree of importance and a low degree of performance belong to low priority, marked as III; those with a high degree of importance and a low degree of performance belong to high priority, marked as IV. One should focus on the fourth quadrant, classify them as key product service touch points, and complete the evaluation of product service touch point importance performance.

### 3.2.3. Analyzing Failure Mode and Effect

After identifying the critical service touch point, an FMEA assessment team was formed by three service designers, three traffic management engineers, and four volunteers to classify and list the failure modes, failure causes, and failure consequences of the key product service touch points. The results of their analysis should be as concise as possible, covering a wide range of possibilities, and be significantly representative. The analysis of failure modes in this stage is the basis for the analysis of emotional interaction factors introduced later.

After the qualitative discussions reached the agreed conclusions, which are listed in the table, the severity ($S$), occurrence ($O$), and detection ($D$) of the product service touch point failure are scored quantitatively by the FMEA team. The higher the value, the more likely the failure is to be dangerous, which means failure is more serious, more likely to happen, or less likely to be detected [41]. The specific scoring standard [42] is shown in Table 1.

### 3.2.4. Clarifying Priority and Redesign

It was at this stage that the topic of emotional interaction was introduced, and it is of great significance to clarify the affective failure risk. In this study, the failure of affective interaction means that passengers have difficulty being emotionally satisfied in speculative design experiments, or are feeling negative emotions such as tension, anxiety, and unease during the experience of the service.

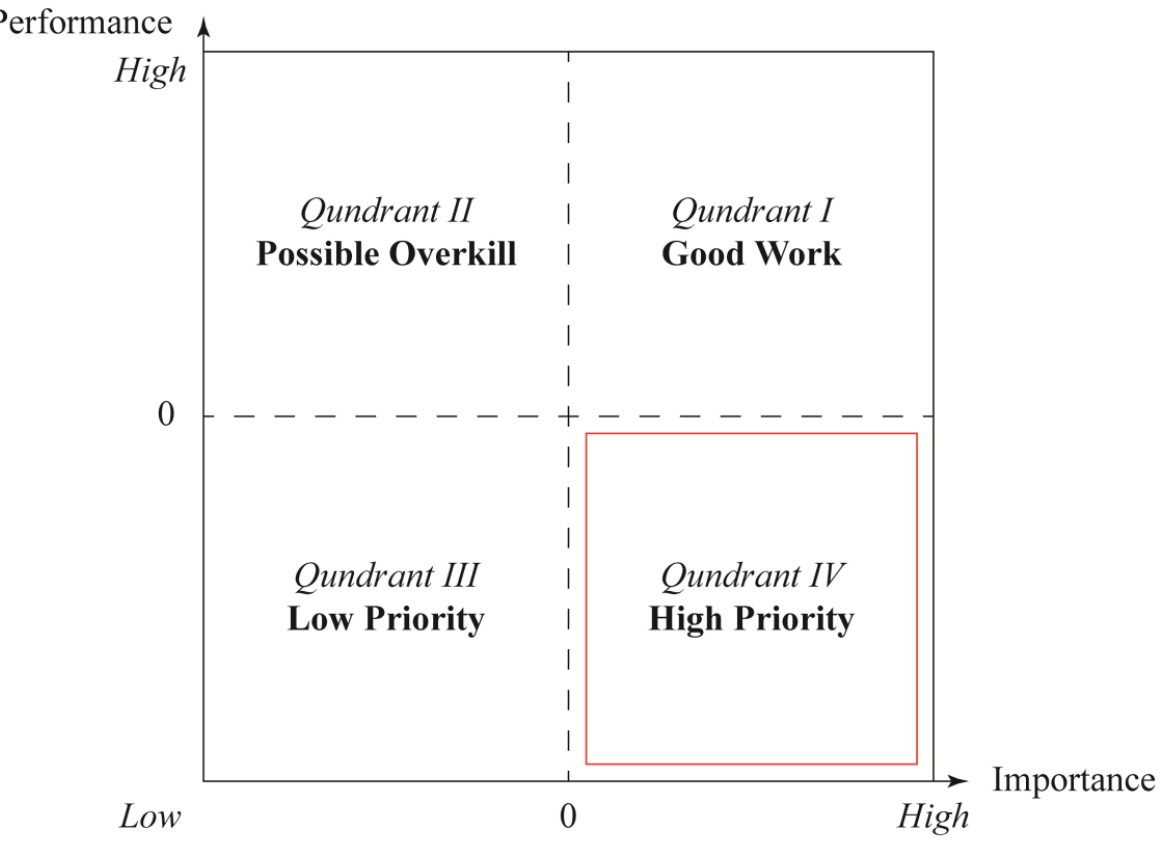

**Figure 5.** The four categories depend on degree of importance and performance.

**Table 1.** Scoring standard of FMEA evaluation.

| Score | Severity | Occurrence | Detection |
|---|---|---|---|
| 9–10 | Great obvious impact; hard to maintain service | Probability > 30% | Non-detectable |
| 7–8 | Huge impact; difficult to maintain service | Probability ≤30% | Experience required |
| 5–6 | Moderate impact, and the service needs to be improved significantly | Probability ≤ 20% | Testing guidelines required |
| 3–4 | Minor impact, and the service needs to be adjusted | Probability ≤ 10% | Expert assessment required |
| 1–2 | No obvious impact | Probability ≤ 1% | Professional assessments and manuals required |

To quantify the effects of failure of emotional interactions, the concept of user-perceived tolerance area is introduced into the FMEA. The user's expected service that affects product design can be divided into two parts: appropriate service and ideal service [43]. Proper service is the lowest service customers expect to receive, while ideal service is the highest service customers expect to receive. The tolerance area is the area formed by the gap between the appropriate service and the ideal service expected by customers. It represents the expected service within the gap range that customers can accept. The tolerance area is determined by the above two expectations, so the appropriate service and the ideal service are measured, and then the tolerance area width is calculated according to the measurement results of the two expectations, as shown in Figure 6 [43].

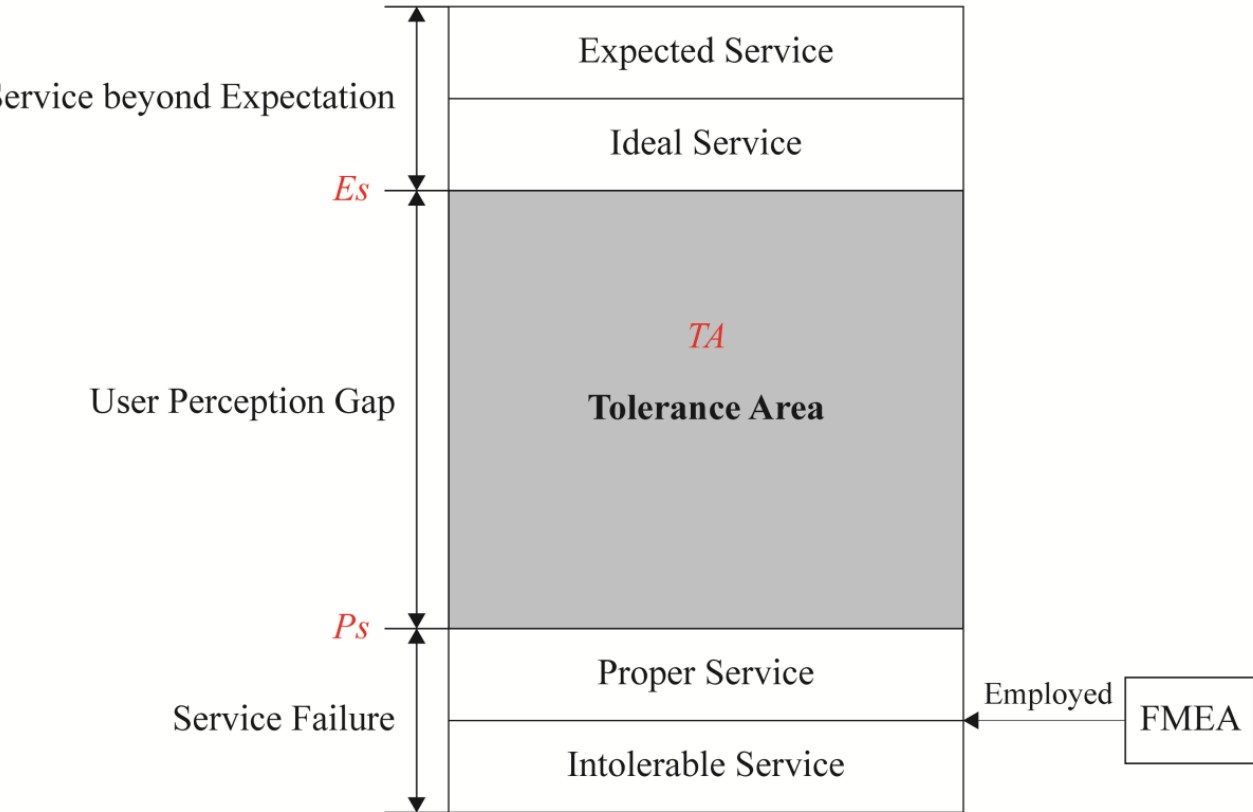

**Figure 6.** Model diagram of service touch point tolerance area.

Assuming that *TA* represents the tolerance area, *Es* represents the ideal service considered by the user, and *Ps* represents the appropriate service value acceptable to the user, as Formula (5) shows. *Es* and *Ps* are integers between 1 and 10.

$$TA = Es - Ps \tag{5}$$

According to the determined failure mode of the key product service touch point above, a second questionnaire is designed to investigate the user expectation and ideal value of driverless ridesharing passengers. Calculating the risk priority factor *ERPN* value of each key product service touch point by Formula (6), the greater the *ERPN* value, the greater the risk of the failure mode, and the more the need to take measures to prevent it [42].

$$ERPN = 3^{W_S \times (S - S_P)} + 3^{W_O \times (O - O_P)} + 3^{W_D \times (D - D_P)} \tag{6}$$

$W_S$, $W_O$, and $W_D$ represent the weight of the ideal service value of the user expectation at the contact time; $S_P$, $D_P$, and $O_P$ represent the average value of the appropriate service score of the severity, occurrence, and detection degree of the user expectation. *S*, *O*, and *D* are the average values of the evaluation value of each key product service touch point. The ideal service value weights $W_S$, $W_O$, and $W_D$ were at the critical value, *ERPN* = 3. When *ERPN* < 3, users can tolerate the current service product service touch point; when *ERPN* > 3, users cannot tolerate it. The higher the *ERPN* is, the higher the priority of redesign is [42]. This stage analyzes the defects of the key risk product service touch point, or the factors causing trouble in a certain part, so as to guide the redesign of the key risk product service touch point [44].

## 4. Research Result

### 4.1. Identifying Service Touch Points

Based on the scenario and theory above, the whole service process can be divided into four stages: deciding and booking (*T1*), consisting of 6 touch points; waiting to be picked up (*T2*), consisting of 9 touch points; traveling and possible stops (*T3*), consisting of 7 touch points; and destination (*T4*), consisting of 4 touch points; 26 touch points in total, as listed in Table 2.

**Table 2.** Service stages and touch points of driverless ridesharing vehicle.

| Stage | Environment | Touch Point | Physical Medium | Code |
|---|---|---|---|---|
| Deciding and Booking (T1) | indoor/outdoor | Download APP | Mobile phone, etc. | T11 |
| | indoor/outdoor | Open APP | Mobile phone, etc. | T12 |
| | indoor/outdoor | Point out the route | Mobile phone, etc. | T13 |
| | indoor/outdoor | Set other requirements | Mobile phone, etc. | T14 |
| | indoor/outdoor | Check order details | Mobile phone, etc. | T15 |
| | indoor/outdoor | Make booking | Mobile phone, etc. | T16 |
| Waiting To Be Picked Up (T2) | outdoor | Find a position to get on | Traffic signs, etc. | T21 |
| | outdoor | Sit or stand to wait | Public chairs, etc. | T22 |
| | outdoor | Check the vehicle | Mobile phone, vehicle license plate, etc. | T23 |
| | outdoor | Check other details | Mobile phone, vehicle profile, etc. | T24 |
| | in-vehicle | Get on | Vehicle door, etc. | T25 |
| | in-vehicle | Choose the seat | Vehicle seats, etc. | T26 |
| | in-vehicle | Put down luggage | Luggage carrier, etc. | T27 |
| | in-vehicle | Sit on | Vehicle seats, etc. | T28 |
| | in-vehicle | Check the route | Mobile phone, etc. | T29 |
| Traveling and Possible Stops (T3) | in-vehicle | Look outside | Vehicle window, etc. | T31 |
| | in-vehicle | Enjoy the scenery | Vehicle fragrance, etc. | T32 |
| | in-vehicle | Listen to music | Mobile phone, earphones, etc. | T33 |
| | in-vehicle | Use mobile phone | Mobile phone, etc. | T34 |
| | in-vehicle | Contact with others | (NONE) | T35 |
| | in-vehicle | Relax | Vehicle seats, etc. | T36 |
| | in-vehicle | Others get off or get on | (NONE) | T37 |
| Destination (T4) | in-vehicle | Check destination | Mobile phone, vehicle window, etc. | T41 |
| | outdoor | Get off | Vehicle door, etc. | T42 |
| | outdoor | Carry luggage | Luggage carrier, etc. | T43 |
| | outdoor | Give feedback | Mobile phone, etc. | T44 |

According to Table 2, the environment of service touch points is divided into three categories: indoor, outdoor, and in-vehicle. The physical medium of service touch points includes parts of the car, such as the car seat, and user terminals, such as mobile phones. Some touch points do not require a physical medium but the atmosphere as an invisible medium.

### 4.2. Importance–Performance Analysis of Touch Points

#### 4.2.1. Reliability Level Analysis

A total of 103 volunteers, including international students, university tutors, and Glasgow citizens, including walkers, bike riders, taxi users, Uber users, bus users, car drivers, and traffic police, were invited to participate in this study. A total of 103 questionnaires were sent out, and 100 valid results were returned. The interviewee group almost covers all roles in transportation, which is highly representative. The data are imported into SPSS, and the reliability test results of the scale are shown in Table 3. The Cronbach's coefficients of the importance and performance of the collected questionnaire data are greater than 0.8, which reaches the level of passing the reliability test, indicating that the reliability of the questionnaire data is high and the data are reliable.

**Table 3.** The Cronbach's coefficients of the importance and performance.

| Objects of Reliability Test | Cronbach's Coefficient | Number of Valid Questionnaires |
|---|---|---|
| Importance Degree | 0.828 | 100 |
| Performance Degree | 0.834 | 100 |

4.2.2. IPA Data Processing and Result

The results of the performance and importance ratings for each touchpoint, as a result of the statistical processing described above, are shown in Table 4.

**Table 4.** IPA result of service touch points.

| Stage | Code | $A_p$ | $S_p$ | $A_i$ | $S_i$ | Number | Category |
|---|---|---|---|---|---|---|---|
| Deciding and Booking (*T1*) | T11 | 3.12 | 0.069268 | 4.49 | 0.255602 | 1 | I |
| | T12 | 2.83 | −0.18758 | 3.81 | −0.35077 | 2 | III |
| | T13 | 3.08 | 0.038297 | 4.03 | −0.22708 | 3 | II |
| | T14 | 3.38 | 0.390855 | 3.94 | −0.38835 | 4 | II |
| | T15 | 3.16 | 0.157974 | 3.8 | −0.52033 | 5 | II |
| | T16 | 3.54 | 0.47388 | 3.71 | −0.49272 | 6 | II |
| Waiting To Be Picked Up (*T2*) | T21 | 2.66 | −0.35646 | 4.48 | 0.186527 | 7 | IV |
| | T22 | 2.57 | −0.39727 | 3.06 | −1.01621 | 8 | III |
| | T23 | 3.26 | 0.201638 | 4.88 | 1.298681 | 9 | I |
| | T24 | 3.25 | 0.204702 | 4.96 | 3.261333 | 10 | I |
| | T25 | 3.16 | 0.127556 | 4.66 | 0.522813 | 11 | I |
| | T26 | 2.90 | −0.09533 | 4.71 | 0.645302 | 12 | IV |
| | T27 | 2.93 | −0.0861 | 4.29 | −0.03247 | 13 | III |
| | T28 | 2.59 | −0.3776 | 4.48 | 0.169329 | 14 | IV |
| | T29 | 3.22 | 0.178075 | 3.35 | −0.75014 | 15 | II |
| Traveling and Possible Stops (*T3*) | T31 | 3.11 | 0.068137 | 4.97 | 2.929067 | 16 | I |
| | T32 | 3.15 | 0.125738 | 4.65 | 0.393093 | 17 | I |
| | T33 | 2.83 | −0.17024 | 4.34 | 0.020225 | 18 | IV |
| | T34 | 2.92 | −0.09173 | 4.41 | 0.118384 | 19 | IV |
| | T35 | 3.01 | −0.0161 | 4.54 | 0.330323 | 20 | IV |
| | T36 | 2.92 | −0.11288 | 4.46 | 0.175843 | 21 | IV |
| | T37 | 2.74 | −0.25348 | 4.7 | 0.594071 | 22 | IV |
| Destination (*T4*) | T41 | 2.89 | −0.1338 | 4.27 | −0.0501 | 23 | III |
| | T42 | 3.22 | 0.158321 | 4.63 | 0.49562 | 24 | I |
| | T43 | 3.02 | −0.00957 | 4.12 | −0.15721 | 25 | III |
| | T44 | 3.31 | 0.298429 | 4.52 | 0.307113 | 26 | I |

The data in Table 4 were plotted according to Section 3.2.2 to facilitate the study, as shown in Figure 7.

The diagram clearly shows that touch points T21, T26, T28, T33, T34, T35, T36, and T37, which are identified as the key product service touch points in Class IV, are of great necessity to identify keys with high failure risk from these eight touch points.

*4.3. FMEA of Key Touch Points*

The failure modes analysis and evaluation results by the FMEA team of 10 members are shown in Table 5.

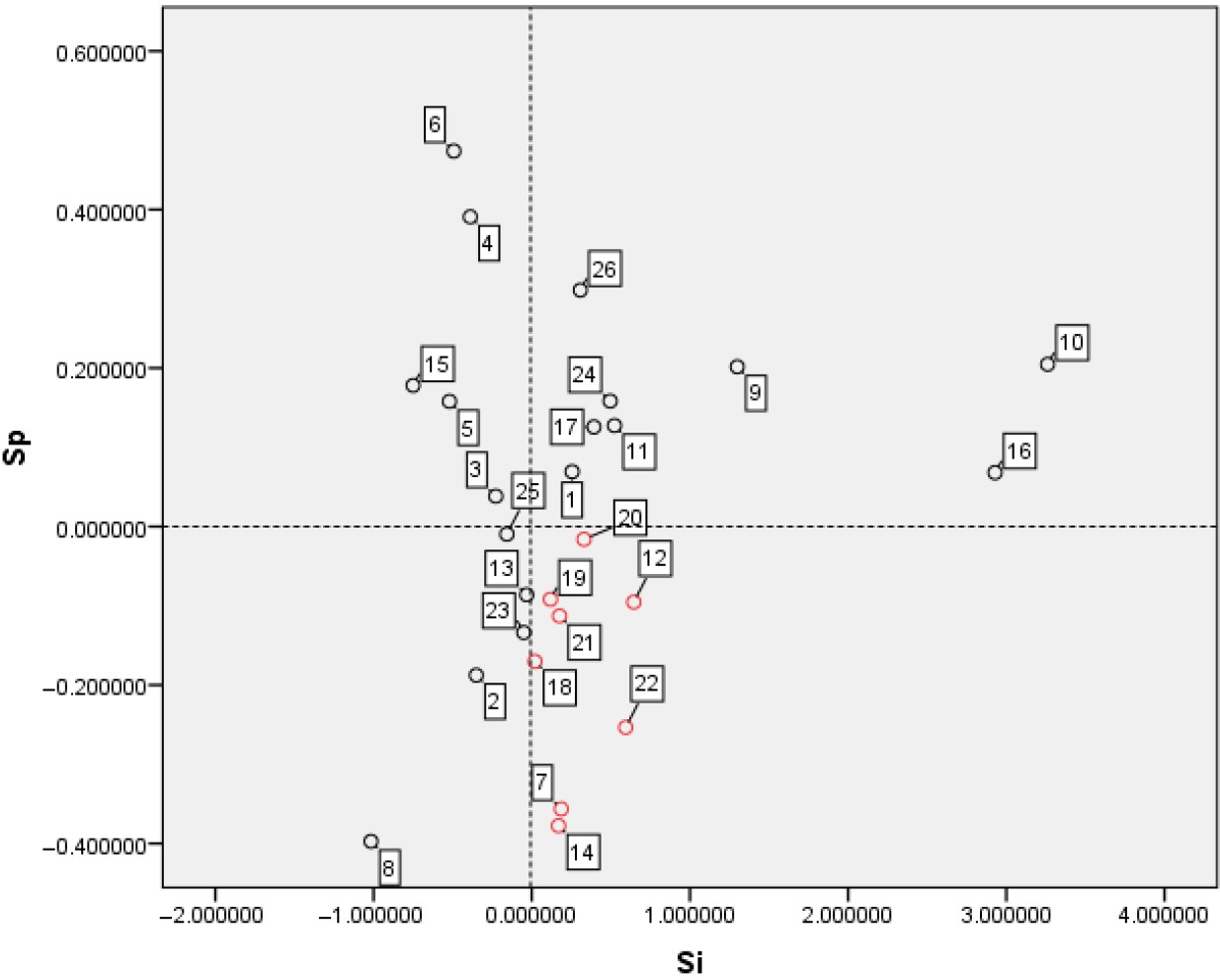

**Figure 7.** Classification diagram of service touch points.

**Table 5.** Analysis result of failure mode evaluation.

| Code | Key Touch Point | Failure Mode | Failure Cause | Failure Consequence |
|------|-----------------|--------------|---------------|---------------------|
| T21 | Find a position to get on | Hard to find an accurate and safe pick-up point | Unmarkable signs and inaccurate navigation | Missing leads to the extra walking |
| T26 | Choose the seat | No preferred seat | Other passengers have chosen | Uncomfortable trip |
| T28 | Sit on | No enough space | Other passengers have occupied | Uncomfortable trip |
| T33 | Listen to music | Disturbed | Noise by other passengers | Less than desirable experience |
| T34 | Use mobile phone | Privacy Crisis | Fear of other passengers seeing the phone | Boring and anxious trip |
| T35 | Contact with others | Embarrassing atmosphere | Eye contact or body contact with other passengers | Embarrassing experience |
| T36 | Relax | Insecurity | Fear of other passengers behaving | Worrying trip |
| T37 | Others get off or get on | Disturbed | Need to give way for others leaving | Uncomfortable trip |

According to Table 5, among these eight key service touch points, the failure of T26, T28, T33, and T37 may cause low emotions of dissatisfaction, and the failure of T34, T35, and T36 likely cause more negative sentiments, such as boredom, embarrassment, anxiety, and insecurity. To be brief, the current result of the FMEA shows that the majority of the key service touch points are related to the emotions of passengers.

The evaluation result of 10 members was aggregated and the arithmetic mean was used to describe the average level of the three criteria. The average of the scoring results for each key touch point is shown in Table 6.

**Table 6.** Failure evaluation result of key touch points.

| Code | Key Touch Point | Failure Mode | Severity (*S*) | Occurrence (*O*) | Detection (*D*) |
|------|-----------------|--------------|----------------|------------------|-----------------|
| T21 | Find position to get on | Hard to find accurate and safe pick-up point | 5.2 | 7.1 | 7.4 |
| T26 | Choose the seat | No preferred seat | 6.9 | 7.5 | 7.8 |
| T28 | Sit on | No enough space | 8.1 | 7.8 | 7.1 |
| T33 | Listen to music | Disturbed | 4.5 | 5.4 | 2.9 |
| T34 | Use mobile phone | Privacy crisis | 5.0 | 3.9 | 1.9 |
| T35 | Contact with others | Embarrassing atmosphere | 7.9 | 8.2 | 8.1 |
| T36 | Relax | Insecurity | 7.7 | 8.0 | 7.0 |
| T37 | Others get off or get on | Disturbed | 6.6 | 6.9 | 6.4 |

According to Table 6, the failure of touch point T28 is the most serious, the failure of touch point T35 is most likely to occur, and the failure of touch point T35 is the most hard to detect. Based on the result in Table 6, by combining the values of the three indicators and analyzing the failure effect of key touch points, further failure risk analysis is supposed to be conducted to select key touch points with high failure risk.

### 4.4. Introducing Affective Interaction into FMEA

#### 4.4.1. Measurement of Tolerance Region

Based on the initial result of the FMEA mentioned above, affective failure is regarded as the main failure mode, and affective interaction is employed as a vital criterion in the second questionnaire. Through the second questionnaire, a total of 20 volunteers from the previous questionnaire interviewee group in Section 4.2.1 were invited to participate in this study, 20 questionnaires were sent out and 20 valid results were returned. These 20 interviewees cover all roles involved in the first questionnaire, which is representative to some extent. The specific evaluation results are shown in Table 7.

**Table 7.** Tolerance region analysis result of key touch points.

| Code | Key Touch Point | *S* | | *O* | | *D* | |
|------|-----------------|------|------|------|------|------|------|
| | | *Es* | *Ps* | *Es* | *Ps* | *Es* | *Ps* |
| T21 | Find position to get on | 1 | 6.2 | 1 | 6.4 | 1 | 5.4 |
| T26 | Choose the seat | 1 | 6.4 | 1 | 5.2 | 1 | 5.0 |
| T28 | Sit on | 1 | 5.0 | 1 | 5.6 | 1 | 4.0 |
| T33 | Listen to music | 1 | 5.4 | 1 | 5.0 | 1 | 3.2 |
| T34 | Use mobile phone | 1 | 5.6 | 1 | 4.8 | 1 | 4.4 |
| T35 | Contact with others | 1 | 6.0 | 1 | 4.6 | 1 | 4.6 |
| T36 | Relax | 1 | 5.7 | 1 | 5.1 | 1 | 4.3 |
| T37 | Others get off or get on | 1 | 5.6 | 1 | 4.0 | 1 | 3.6 |

The average value of the appropriate service score of the severity, occurrence, and detection of user expectation is expressed as *SP*, *DP*, and *OP*. By calculating the data in the table, we can obtain the appropriate service scores of the user's expected severity, occurrence, and detection at each critical contact time; *SP*, *DP*, and *OP* were 5.74, 5.08, and 4.31, rounded to 6, 5, and 4. The ideal service value is 1.

#### 4.4.2. The Priority of Failure Risk

Based on the appropriate service values of the above key user severity and occurrence measures, assuming that the weights of severity, occurrence, and detection are the same,

that is, the values of $W_S$, $W_O$, and $W_D$ are all 1, then Formula (6) becomes Formula (7) [44], and the analysis result by Formula (7) is listed in Table 8.

$$ERPN = 3^{S-6} + 3^{O-5} + 3^{D-4} \tag{7}$$

**Table 8.** Failure risk analysis result and priority.

| Code | Key Touch Point | S | O | D | ERPN | Failure Risk |
|------|-----------------|-----|-----|-----|--------|--------------|
| T21 | Find position to get on | 5.2 | 7.1 | 7.4 | 52.36 | 5th |
| T26 | Choose the seat | 6.9 | 7.5 | 7.8 | 83.29 | 2nd |
| T28 | Sit on | 8.1 | 7.8 | 7.1 | 61.85 | 3rd |
| T33 | Listen to music | 4.5 | 5.4 | 2.9 | 2.04 | 7th |
| T34 | Use mobile phone | 5.0 | 3.9 | 1.9 | 0.73 | 8th |
| T35 | Contact with others | 7.9 | 8.2 | 8.1 | 132.10 | 1st |
| T36 | Relax | 7.7 | 8.0 | 7.0 | 60.47 | 4th |
| T37 | Others get off or get on | 6.6 | 6.9 | 6.4 | 23.96 | 6th |

According to Table 8, the touch points with a high risk of failure are concentrated in stages T2 and T3. T35 has the highest risk of failure, which deserves more attention; T26 and T28 in stage T2 are in second and third place, followed by T36, T21, and T37. T33 and T34 have an *ERPN* value of less than 3, indicating that they have the lowest risk of failure and are of lower priority for a redesign.

*4.5. Redesigning for Touch Point with High Failure Risk*
4.5.1. Analyzing the Essential Causes of Failure

These touch points with a high risk of failure can be divided into two categories according to the factors that influence the affective interaction of the users. The first category is influenced by tangible physical material and the other is influenced by immaterial social distance and atmosphere. T21, T26, and T28 belong to the former category. T33, T34, T35, T36, and T37 belong to the latter category.

In terms of tangible material, the priority of T26 and T28 is supposed to be higher because of the higher *ERPN* value; furthermore, both two touch points are related to seats in vehicle. The main failure cause of T26 is that the passengers that get on early might occupy their preferred seats, which means that the passengers that get on later cannot choose their preferred seats. The seat is the primary medium of interaction during traveling, and the position of the seat has a huge impact on the emotions of the passengers. The main failure cause of T28 is that the vehicle, especially the seats inside, provides such limited space for passengers that it is easy for them to get caught up in tension, anxiety, and unease. [45] The medium for both touch points is the seat, and therefore, the seat should be considered the main object for a redesign. Reducing the differences between the individual seats in vehicles and providing more personal space for individual passengers are the main areas of optimization [46].

In terms of intangible atmosphere, the main failure causes of T33, T34, T35, T36, and T37 are all related to contact with unfamiliar strangers. The essence of these reasons is fear of socializing with strangers; in other words, insufficient social distance [45]. In terms of shared transport, deeper findings show that the so-called convenience, affordability, efficiency, and environmental friendliness that come with sharing are of less importance in the views of passengers than privacy and social distance [46]. The most remarkable pain point found after the FMEA is related to privacy and social distance, which suggests that redesigning a more comfortable affective space can provide a better ridesharing service, such as reorganizing seats to offer more independent seating spaces.

To summarize, the majority of key touch points with high failure risk are related to social phobias. From the perspective of affective interaction, the core requirements of passengers are sufficient social distance, as little interpersonal contact as possible, and a quiet atmosphere in the vehicle [45,46], which the initial prototype is not able to offer. Affective interaction is, therefore, a key focus of the redesign, and the new speculative

design prototype should allow the service to meet the affective needs of the passenger and allow the passenger to feel as much affective feedback as possible.

### 4.5.2. Targeted Design Decision

In response to the results of the above analysis of the failure modes and causes, several possible design decisions are presented, as shown in Figure 8. The core of the solution is the redesign of the seats in the vehicle to achieve almost no difference between each seat, while providing flexible and controlled privacy for each passenger. The seat has a rotatable semi-enclosed capsule type design, and it can be controlled by the end-user device, which not only provides a limited range to avoid contact with strangers, but the optional orientation also further prevents embarrassing social scenarios. The boxy body with the larger doors and the 2×3 seats arrangement equates to more interior space.

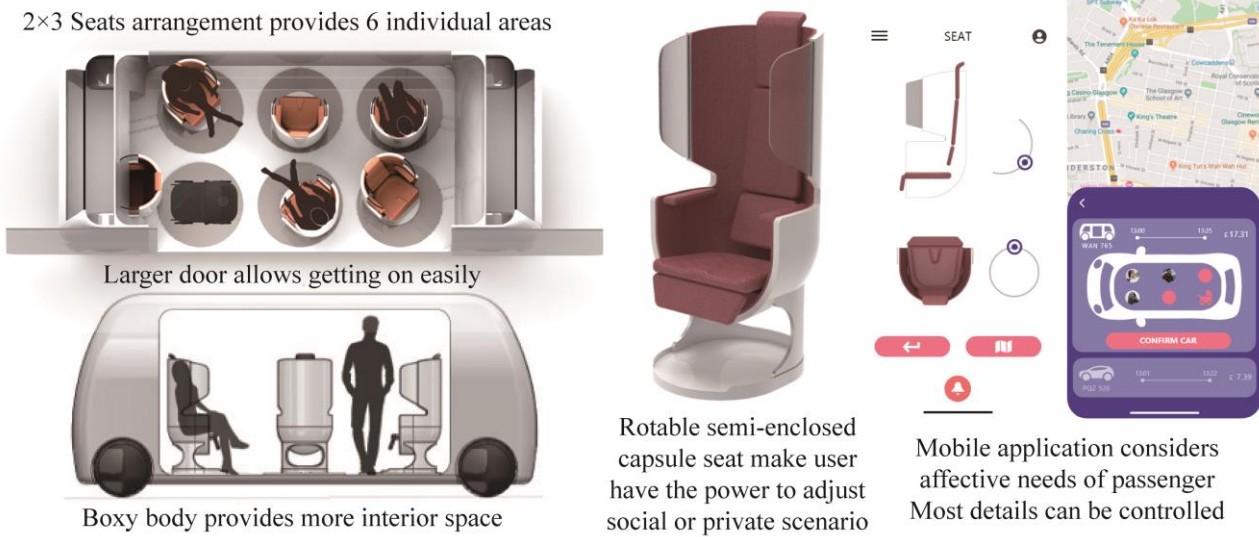

**Figure 8.** Targeted design decisions of driverless ridesharing vehicle.

Among the above-mentioned measures, the majority belong to the improvement and optimization of the original service touch points, as follows: the larger box-shaped carriage allows passengers to have plenty of space while avoiding contact with other passengers in the middle of the journey; larger doors on both sides make it easy for passengers to reach each seat without disturbing others; the rotatable capsule seats block some of the unnecessary social range, while giving passengers the ability to choose their orientation, making the social environment flexible and controllable; the increased interior space resulting from the reshaping of the car body, the reduced difficulty of boarding due to the enlargement of the doors, and the formation of social barriers due to the redesign of the seats are all optimizations in the form; the redesigned physical medium can serve to a certain extent to create a preferred social atmosphere for the passengers in the speculative vehicles to achieve an intangible and quality affective interaction.

Some of the solutions create a new service touch point. As mentioned above, the rotatable seats provide passengers the ability to adjust the orientation; the seat control becomes the new service touch point, whose environment is in-vehicle and the physical media are the mobile application and seat, and its main aim is to be able to accommodate the need for passengers to adjust their orientation. After the passengers adjusted to their preferred orientation, complemented by the semi-enclosed shell of the seats, the interior space is flexibly divided into individual spaces so that the affective atmosphere of most individual passengers can be met. Even for a small group of two or three passengers, the controllable seats allow them to choose to face each other without disturbing others, which can be regarded as a flexible affective interaction.

## 5. Discussion

### 5.1. Methodology Applied

The core aim of this research is to identify worth in the main aspect of great redesigning from the complex service process of public ridesharing, and to explore the factors that influence the user experience of mini public transport services, where three main analyzing methods and one core analyzing criterion were employed. Identifying service touch points, IPA, and FMEA, contributes to finding insights into complex service processes, which means that the problem of identifying the vital contradictions in service redesign is solved. With the introduction of affective interaction as the main failure criterion [47], the analysis of key touch points is carried out in a direction that focuses more on the emotional value of the user, and the results are highly informative for analyzing failure in service processes [33], which means that the problem of identifying the core aspects of vital contradictions is solved [48].

In detail, four methods were used in four stages. Service touch point analysis provides a tool to define basic individual units and modularize the complex service process. IPA is the first step to identifying key factors by setting importance and performance as two principles. FMEA provides a tool to distinguish the most impactful form of failure, which is the affective failure. Therefore, the last step is to select key touch points with high failure risk by regarding affective interaction as the most possible failure cause. The method system of these four applied methods is best named IPA-FMEA. Furthermore, this method is also suitable for other product design processes [35,36], especially for the service process with more product service touch points in the service process [42]. This method can identify the failure risk of product service touch points more quickly, and then improve the service quality.

In terms of the specific measure, this paper involved a speculative design experiment, two questionnaires, scoring, a team evaluation and discussion, and two quantitative analyses. The highlights among these may be the application of new variables in quantitative IPA analyses and applying the measurement of the affective failure tolerance region. Replacing the mean value of importance and performance with the standard score can be best characterized by a possible decrease in sample errors and systematic errors [39], which seems more objective and quantitative [40]. The indicator of affective failure tolerance may play a constructive role in quantitative estimating failure risk of key service touch points and in distinguishing the most deserving prevention from affective failure [49].

### 5.2. Interpersonal Affective Failure as the Main Mode

According to the final results, reluctant interpersonal communication seems to be the main affective failure mode with a high risk of service touch points. The majority of touch points related to interpersonal interaction and social distance show a high level of importance and a low degree of performance by IPA and a high risk of failure by affective FMEA. Complementary participant interviews indicate that more passengers required privacy, adequate space, and a sense of security in mini public transport [50], which might be seen to some extent as a type of social phobia [51]. Therefore, the most serious failure may be the affective failure, especially interpersonal affective failure, which means that passengers have difficulty being emotionally satisfied.

It would be feasible to try and generalize the above argument to other similar public transportation. In principle, the impact of social phobia on passenger service experience is similar in public places [50], so in practice, avoiding an awkward social atmosphere deserves to be taken seriously [52]. The core result and extension of this research is the identified main contradiction in the service process of public transport is interpersonal affective interaction. Since sustainable development requires the promotion of public transportation and one of the main measures to promote it is to improve the service experience, it is of great importance to focus on providing a preferred environment for interpersonal communication for all modes of public transportation [53], which emphasizes the important role of analyzing affective failure [47]. In addition to ridesharing, the

interior spaces of public transport, such as minibuses, carriages of trains, and rooms on ships, are supposed to be considered by introducing interpersonal interaction as the main affective factor [51]. Public transportation that is friendly to social phobias should be more advantageous [49]. As for the specific forms, specific modes of transport need to be analyzed individually, but this paper does not draw on their design.

The insightful finding that interpersonal affective failure is a vital aspects affecting the ridesharing service has been proposed before. A Chinese ridesharing market research report cited the awkward emotional atmosphere inside vehicles as one of the three main factors deterring passengers from using public transport; the other two being the longer commute time and strong concerns about personal safety [54]. Research from other groups found similar conclusions [55]. Companies in China have, therefore, strengthened their policies on the supervision of vehicles and installed cameras inside them to avoid possible accidents [52]. Passengers themselves also take spontaneous steps to avoid unnecessary social interaction. However, these initiatives do not address the potential emotional failure of passengers, and as service providers, platforms do not currently have an effective solution [54]. Moreover, some studies from other countries have also found the importance of emotional interaction, but have not paid sufficient attention to it [51,52,54], which is their biggest difference with the past related literature.

### 5.3. Reliability of the Research Result

The results of this study are highly representative and reliable because the research scenario and model can represent mainstream ridesharing services and the research participants can represent public transport participants. The prototypes built through speculative design are the generalization of several similar applications [14,15], which means that the service logic and process are most commonly used in the current market and represent more practical commercial products. The research scenario and prototype, as the foundation of the research, are so representative that the result may be reliable to a larger extent and may help avoid some uncertainty.

Taking the uncertainty of participants into consideration, occupations or roles and the reliability of the questionnaire are important. Almost every role in public transport, such as walkers, bike riders, taxi users, Uber users, bus users, car drivers, and traffic police, was represented in the respondent sample. As for questionnaire reliability, Cronbach's coefficients in Section 4.2.1 prove that the first questionnaire is reliable. Additionally, a statistical significance test using a threshold of $\alpha = 5\%$ aimed to evaluate whether the interviewees of the second questionnaire could represent the interviewees of the first questionnaire. The confidence levels of 25 touch points are 95% and only one touch point's significance level is approximately equal to 10%, which means that the second respondent group basically reflects the characteristics of the first respondent group [56]. Both the qualitative and quantitative analyses above confirm the reliability of the findings.

### 5.4. Contributions to Knowledge and Practice

To summarize, the contribution of this work mainly contains two aspects, the importance of interpersonal affective interaction in public transport and a new approach to design decisions based on IPA-FMEA.

In combination with the findings of other researchers [54,55], regardless of the form of public transport, it should be common sense and a universal standard to emphasize the issue of passenger privacy and interpersonal affective interaction while providing services.

In other design practices, the design decision method based on IPA-FMEA in this study is beneficial with the introduction of the standard score, but not the mean value and the consideration of affective failure tolerance in clarifying the priority of numerous detailed points [51,52,54].

*5.5. Limitations and Further Work*

All experiments were conducted using a speculative design approach through simulated immersive experiences, and although the sample of 100 participants should be highly representative, the subjective perceptions of the volunteers may still be limited [14,15]. The collection of user perceptions, mainly in the form of questionnaires and the conversion of subjective perceptions into quantitative scores through statistical methods, does not, to a certain extent, avoid the possible influence of subjective factors, such as stereotypes and biases, on the results.

The further speculative design experiment should be tested and the subsequent speculative prototypes are supposed to be analyzed by IPA-FMEA to verify the reasonableness and feasibility of this research conclusion again. On the other hand, this method requires a large amount of calculation, but the calculation model should be applied universally, and the calculation accuracy should be improved.

## 6. Conclusions

Due to the social benefits of improved service experience for the promotion of sustainable public transport, this research focuses on the affective interaction of in-vehicle users, which has not been thoroughly studied. Based on speculative design experiments, a novel approach, IPA-FMEA, is utilized to investigate the impact of each service touch point on the passenger experience. The findings suggest that reluctant interpersonal communication seems to be the main affective failure caused by the high risk of service touch points. Public transportation that is friendly to social phobia should be more advantageous. Despite some limitations, the information reported in this study may help to better design mini public transport service systems and create better emotional interaction environments to meet the needs of passengers. Understanding the preferences of the public and emotional perceptions of the interpersonal interaction environment could improve the design of public transport service processes, taking into account their psychological and sustainable advantages to facilitate more people choosing public transport.

**Author Contributions:** Conceptualization, methodology, and validation, Q.P. and W.W.; data software, data curation, Q.P. and Y.W.; formal analysis, writing, Q.P. and X.Y.; writing—review and editing, W.W. and J.C. All authors have read and agreed to the published version of the manuscript.

**Funding:** This work was supported by the Innovation Capability Support Program of Shaanxi (No. 2021PT-025), the Key R&D Plan Program of Shaanxi (No. 2022ZDLGY06-05), the "Young outstanding" Talent Support Program of Colleges and Universities in Shaanxi (No. 2020-50), and the Research Program of Humanities and Social Sciences of Shaanxi Provincial Department of Education (No. 21JK0070).

**Informed Consent Statement:** Informed consent was obtained from all subjects involved in the study.

**Data Availability Statement:** The data are not publicly available due to the ongoing research, and the authors will continue to work with it in the future.

**Conflicts of Interest:** The authors declare no conflict of interest.

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
