# Peer review of "Research on Affective Interaction in Mini Public Transport Based on IPA-FMEA"

_sustainability, doi:10.3390/su15097033_

Round 1
Reviewer 1 Report
Authors presented a product service touch point 9 evaluation approach which based on the importance-performance analysis (IPA) of user and failure 10 mode and effect analysis (FMEA). Here are some comments which should be addressed before publishing.
The statistical significance discussion is missing. How the uncertainty in responses was taken into consideration? The responses number is quite low, please introduce a support during discussion from literature. The authors should improve the analysis to support learnings.
Reviewer 2 Report
This research is well done. Only one needs strengthening. The supplement of literature needs to be improved, this includes a literature review, discussion, and conclusion. This is a great lack of theory for readers. In particular, this research is not completely absent in the past. This is because the author must carefully consider the rationality, gaps, and appropriateness of the theory.
1. Whether the research method is in line with the context of the research topic should be able to provide relevant theoretical discussions as the core theory supporting this research
2. Discussion should add manuscripts to respond to the deficiencies in the manuscripts of this research improvement, such as the failure of the service process, why can the blindfolds be untied, and why did these blindfolds arrive?
3. There is a lack of literature to support the results, so the results of the research and the cost of the research are all imaginary. Along with discussions of these papers, it would appear that this study is too assertive for the needs of an academic audience.
4. Furthermore, the research results should find differences with the past related literature, but the author did not bring it up, which will also make readers think that the author has ignored the conclusions of related theories.
Reviewer 3 Report
In the manuscript titled Research on Affective Interaction in Mini-Public Transport Based on IPA-FMEA , the paper captures service product service touch points and perform the correlation analysis of it.The author presents a product service touch point evaluation approach which based on the importance-performance analysis (IPA) of user and failure mode and effect analysis (FMEA) and apply it to the experiment. Their target is to examine the main factors affecting the passenger experience of mini-public transport services and further analyze their manifestations, main consequences and risks of failure, based on which recommendations for improvement and optimization are made to avoid the creation of pain points.
1.This study contains some interesting findings and are valuable for the understanding of analyze the failure of key product service touch points based on user perceived affective interaction, and clarify the priority of each key touch point.
2. ln the introduction section, the authors need to determine the specific application scenarios and use data to show why the problem is important, how current solution are inadequate, and why the problem is difficult to solve.
3.However, lack of comparison with other evaluation methods and case analysis is the major flaw of the study.
4. The data in the research are not sufficient to demonstrate the result.In the 4.4 section. It is not stated whether the experimental samples come from which social groups, if it is the same social group, the analysis results are too one-sided.
5. The current manuscript needs to be polished by a native English speaker or a professional language editing service.
Round 2
Reviewer 1 Report
The article is much improved. Please correct grammatical mistakes and proofread by native speaker.
Reviewer 3 Report
Thanks for the efforts of responding my comments. You have addressed my concerns to a satisfactory level.